# Phase Transitions of Cu and Fe at Multiscales in an Additively Manufactured Cu–Fe Alloy under High-Pressure

**DOI:** 10.3390/nano12091514

**Published:** 2022-04-29

**Authors:** Arya Chatterjee, Dmitry Popov, Nenad Velisavljevic, Amit Misra

**Affiliations:** 1Department of Materials Science and Engineering, University of Michigan, Ann Arbor, MI 48109, USA; 2Argonne National Laboratory, HPCAT, X-ray Science Division, Lemont, IL 60439, USA; dpopov@anl.gov (D.P.); hpcat-director@anl.gov (N.V.); 3Lawrence Livermore National Laboratory, Physics Division, Livermore, CA 94550, USA; 4Department of Mechanical Engineering, University of Michigan, Ann Arbor, MI 48109, USA

**Keywords:** Cu–Fe alloys, additive manufacturing, hierarchical microstructure, phase transformation, Cu and Fe precipitates

## Abstract

A state of the art, custom-built direct-metal deposition (DMD)-based additive manufacturing (AM) system at the University of Michigan was used to manufacture 50Cu–50Fe alloy with tailored properties for use in high strain/deformation environments. Subsequently, we performed preliminary high-pressure compression experiments to investigate the structural stability and deformation of this material. Our work shows that the alpha (BCC) phase of Fe is stable up to ~16 GPa before reversibly transforming to HCP, which is at least a few GPa higher than pure bulk Fe material. Furthermore, we observed evidence of a transition of Cu nano-precipitates in Fe from the well-known FCC structure to a metastable BCC phase, which has only been predicted via density functional calculations. Finally, the metastable FCC Fe nano-precipitates within the Cu grains show a modulated nano-twinned structure induced by high-pressure deformation. The results from this work demonstrate the opportunity in AM application for tailored functional materials and extreme stress/deformation applications.

## 1. Introduction

Laser-based additive manufacturing techniques such as powder-bed or directed powder deposition have attracted interest in recent years as effective tools to build 3D-printed metallic alloys [1,2,3,4]. The microstructures produced are significantly different than that produced with the conventional route owing to the rapid heating and cooling associated with additive manufacturing (AM) using a focused laser beam, as well as variation in other process parameters, e.g., size and shape of localized melt pool and scanning speed and pattern [1,2,3,4]. Repeated deposition of newly molten layers during AM causes high thermal gradient and temperature variations that lead to the segregation of constituent elements, evolution of inhomogeneous grain morphology, dislocation substructure and crystallographic textures [1,2,5]. Extensive amounts of research works on AM of metallic systems have been primarily directed to the commercial austenitic stainless steels, and Al-, Ni- and Ti-base alloys, and some advanced materials such as high entropy alloys, bulk metallic glasses, metal matrix composites and Mg alloys [6]. In comparison, there are very few research works on AM of Cu-rich alloys [7,8,9] since the high reflectivity of Cu makes it difficult to create a stable melt pool using a focused laser beam. Thus, there is a need for in-depth studies of its microstructure evolution and, in particular, stability under extreme pressure, temperature and high-strain-rate conditions, etc. At present time, based on available literature, there are no reported studies that have investigated the microstructural stability of AM Cu–Fe alloy under high-pressure conditions.

In the current study, an equiatomic AM Cu–Fe alloy (50Cu–50Fe) with a hierarchical microstructure, i.e., phase-separated grains of Cu and Fe as well as the presence of Cu precipitates in Fe grains and Fe precipitates in Cu grains, is subjected to very high hydrostatic pressure. At ambient conditions, pure Fe metal exists in a body-centered-cubic (BCC) crystal structure (α phase). At high pressures, Fe is known to transition into the ε phase with a hexagonal-closed-packed (HCP) lattice. This pressure-induced phase transition of *α*-Fe (BCC) → *ε*-Fe (HCP) is previously reported as martensitic in nature and occurs at approximately 13 GPa pressure [10]. Earlier works have shown that the phase transition of α-Fe (BCC) to *ε*-Fe (HCP) is a first-order phase transformation and athermal in character [11,12,13,14]. Several studies had exhibited the transformation of *α*-Fe to *ε*-Fe under static [15,16,17] as well as dynamic [18,19,20,21,22] compression conditions. Most of these investigations [15,16,17,18,19,20,21] indicated that this mechanism of phase transition of *α*-Fe to *ε*-Fe followed Burgers path, which was originally proposed by Burgers referring to a *α*-Zr↔ *β*-Zr transformation [23]. However, apart from the phase transition of *α*-Fe to *ε*-Fe, for a certain range of pressure and temperature, α-Fe is found to be transformed into reversible *γ*-Fe with a face-centered cubic (FCC) structure [24,25,26]. Although *γ*-Fe with an FCC structure is stable at higher temperatures, in certain instances, this can be found to be present at ambient condition as well, especially during solidification process in Cu–Fe alloys [27,28,29]. Similar to the *α*-Fe, *γ*-Fe has also been found to undergo martensitic phase transformation without being aided by higher temperatures owing to the application of deformation [30,31]. Earlier works showed the creation of banded structures in *γ*-Fe particles owing to the interaction with glide dislocations and/or formation of twins [30,31,32,33].

In recent years, the martensitic phase transformation of Cu has attracted great attention. This phenomenon is important to the current context as the material in this study contains FCC Cu in the form of microscale grains as well as nanoscale precipitates. The FCC lattice structure of Cu is the stable state, with some indication from first-principles energy calculations that the BCC and HCP Cu could be expected [34,35]. Earlier experimental works showed the presence of high-energy Cu phases in the form of BCC and HCP Cu in the pseudomorphic Cu films deposited on the {001} surface of Fe [36,37] forming multilayers of Cu with Nb [38,39,40], Pt [41], Pd [42] and Ag [43]. The path for the phase transformation of Cu into its higher energy metastable phases such as BCC Cu and 9R Cu has been studied [44,45,46], and it has been found that these phases can be stabilized in the regions of extended defects by imposing constraints using interfaces such as grain boundaries [44]; on the other hand, HCP Cu was found to be stable in films of Cu on {001} W substrate in very small regions even in films thicker than 20 Å [47]. Molecular dynamics simulation revealed that the BCC structure of Cu is elastically unstable, while the HCP and 9R structures of Cu are metastable in nature [35]. However, to the best of the authors’ knowledge, the phase transition of both Cu and Fe in an additively manufactured microstructure subjected to high pressures has not been studied before. In the present study, we report for the first time the investigation of the phase transition of both Cu and Fe at different length scale domains in an additively manufactured equiatomic Cu–Fe alloy under high-pressure.

## 2. Experimental Procedure

### 2.1. Material Preparation

In the current investigation, an equiatomic Cu–Fe alloy (i.e., 50Cu–50Fe) made by direct-metal deposition (DMD)-based additive manufacturing technique was used. Gas-atomized Cu (particle size distribution: d5–d75 µm) and Fe powders (particle size distribution: d25–d150 µm) with nominal composition of 99.999% Cu (Cu-159-1, Praxair Inc., Danbury, CT, USA) and 99+% Fe, 0.05% Oxygen, 0.006% Sulfur and 0.001% Phosphorus (IRON195SP, MiniScience Inc., Main Avenue Clifton, NJ, USA), respectively, were used for the fabrication of this alloy. The substrate used for deposition of composites was an uncoated mild steel 1/4 inch thick. A custom-built DMD system at the University of Michigan was operated using a 4 kW Nd:YAG laser unit with a laser beam diameter of 1 mm to manufacture the 50Cu–50Fe alloy. Argon was used as a shielding gas to avoid oxidation and as powder carrier. More details of processing-related information can be found in our previous publication on the same material [7].

### 2.2. In Situ Characterization of Phase Transition by Synchrotron Radiation

Samples on the order of 50 × 50 × 30 μm^3^ were cut from the bulk of the 50Cu–50Fe sample using the laser cutting system available at the High Pressure Collaborative Access Team (HPCAT) [48] (Figure 1a,b). Using the micromanipulator available at the HPCAT the samples were put into diamond anvil cells (DACs). Ne was used as pressure transmitting medium. In situ X-ray microdiffraction measurements with monochromatic beam focused down to 3.5–6 μm were conducted at 16BMD [49] and 16IDB [50] beamlines of Advanced Photon Source (APS) at Argonne National Laboratory. As Laue diffraction data can be collected much faster than with the monochromatic beam diffraction technique, in situ white beam Laue X-ray diffraction measurements were performed by using the experimental setup available at 16BMB beamline of APS [51]. X-ray polychromatic (white) beam, having the highest energy limit of 90 keV, was focused down to 2 μm^2^. Series of 2D translation Laue diffraction scans, with step size of 2 μm, were collected in situ along with increase in pressure between 5.3 and 18.3 GPa, as well upon decompression (i.e., recovery of sample back to ambient pressure conditions). For the Laue diffraction measurements, special care was taken to avoid “bridging” of the sample between diamond anvils and keep compression hydrostatic across the sample. After completing the Laue measurements, the sample was further studied at 16IDB at highest pressure and using a monochromatic beam, which allowed further identification of the crystalline phases present in the matrix areas and the obtaining of information on relevant sample texture. Series of angular scans with step size of 1° were collected across the samples with monochromatic beam during sample compression up to highest pressures and after full decompression.

As the laser cutting procedure may affect the microstructure of the samples, Laue diffraction data from the bulk samples of Cu50Fe50 were also collected for comparison. The bulk sample blocks were parallelepiped-like shaped with sizes of approximately 1 × 1 × 0.5 mm. DAC-sized samples were cut from the areas near the edges of bulk samples as presented in Figure 1, and the 2D translation Laue diffraction scan was collected over this area in step size of 2 μm. All of the polychromatic beam diffraction measurements were conducted in transmission geometry with the X-ray beam having the highest energy at about 90 keV and focused down to 2 μm^2^. The area detector was tilted vertically by 30° with respect to the incident beam and positioned at ~600 mm from the sample.

During the in situ Laue diffraction experiment, pressure was increased remotely using a membrane system. Pressures before and after collection of Laue data were measured using an off-line Ruby system [52] available in the experimental station. Pressure was also measured simultaneously during the collection of Laue diffraction data with gold standard using a monochromatic beam switchable with the white beam. Analysis of Laue data including indexation and mapping of reflections was performed with software polyLaue developed by Popov et al. [51]. Dioptas [53] X-ray diffraction conversion program was used to integrate diffraction patterns obtained with monochromatic beam and to visualize Laue diffraction patterns, while the Fit2d program [54] was used to find positions of Laue reflections on X-ray images and visualize maps of reflections.

### 2.3. Ex Situ Characterization of Phase Transition by Electron Microscopy

Following the high-pressure compression study, the pressure was released, and the recovered sample was still contained inside the gasket assembly used in the DAC experiments (Figure 2a,b). Further, these were inserted into a Thermo Fisher Helios 650 nanolab dual beam scanning electron microscope (SEM) coupled with focused ion beam (FIB) for preliminary microscopic assessment of the deformed samples as well as for preparing samples for transmission electron microscopy (TEM). TEM specimens were directly prepared from the pressured samples using FIB with a Ga+ ion source. For thin slice preparation, initial rough cuts were at operating conditions of 30 kV, 1 nA, followed by fine thinning at 30 kV and 100 pA. Finally, the electron transparent thin conditions were achieved by operating at 30 kV and 7 pA. Subsequently, ex situ TEM studies of the post-deformed samples were performed with high-resolution electron microscopy in high-angle annular dark field (i.e., HAADF) mode under STEM condition using a double-Cs corrected JEOL 3100R05 TEM, operated at 300 kV.

## 3. Results

### 3.1. Phase Transition at Micrometer Length Scales

The representative micrographs show the presence of two phase-separated regions of Cu and Fe grains; Figure 3a. Moreover, the presence of precipitates was also seen in Cu and Fe grains, i.e., Cu precipitates in Fe and Fe precipitates in Cu grains; Figure 3a. Detailed characterization of these precipitates showed the coherent nature of Fe precipitates (having FCC structure) in Cu grains, whereas the FCC Cu precipitates follow semi-coherent Kurdjumov–Sachs (KS) orientation relationship with the adjacent Fe grains; Figure 3b,c.

The formation of a hierarchical microstructure in typically immiscible Cu–Fe alloy can be attributed to the high cooling rate associated with the laser direct metal deposition (DMD) additive manufacturing technique. The solid solubility of Cu in BCC Fe above 600 °C is very low (the highest being 1.9 wt.% at 850 °C) and almost negligible below this temperature even under equilibrium condition [55]. The very fast cooling rate associated with the DMD process results in entrapment of Cu in Fe grains during the initial stage of solidification. During subsequent deposition, the already-deposited layer adjacent to the laser scan melts, and during this time the supersaturated Cu is rejected from the primary Fe dendrite nuclei into the melt, producing solute partitioning into the liquid ahead of the solidification front. This solute enrichment causes a corresponding variation in the liquidus temperature leading to higher constitutional supercooling (CS). As the solidification progresses, CS causes instability at the solid–liquid interface and provides an additional driving force for the nucleation of more fine Fe grains ahead of the solid–liquid interface.

Once a nucleus is formed, its growth is influenced by kinetics of atom attachment to the interface, capillarity, and diffusion of heat and mass at the interface and away from the interface. The rejected Cu at the solid–liquid interface is likely to influence these factors and restrict the growth of existing Fe nuclei. The excess Cu rejected from the solid will accumulate in an enriched boundary layer ahead of the interface [56]. Possibly at Cu concentrations below a critical value, the rejected Cu ahead of the interface can be dispersed readily into the remaining liquid and significant grain refinement could not be achieved, as we observed in low Cu concentration alloy, i.e., in Cu25Fe75 [7]. When a critical amount of Cu (possibly 50% is sufficient for that) is present in the melt, the Cu-rich layer retards the growth of the nuclei, thereby allowing more Fe nuclei to form in the surrounding super-cooled melt, leading to a fine grain size for Fe grains.

The nano-precipitates observed within both the constituent phases in DMD alloy have not been reported in Cu–Fe alloys processed by other techniques. The solidification rates associated with laser additive manufacturing are quite high, typically >1000 K/s [57]. During the DMD process, each deposited layer of material inadvertently experiences a number of post-deposition re-heating cycles during the deposition of subsequent layers on its top. Elemental Cu and Fe have negligible equilibrium solid-state solubility in Fe and Cu, respectively. Therefore, it is plausible that while the rapid solidification rates could result in a supersaturation of Cu in Fe grains and Fe in Cu matrices, the subsequent post-deposition re-heating cycles will lead to solid-state decomposition of these supersaturated regions. Such decomposition results in the formation of a homogeneous and highly refined distribution of nanoscale Fe precipitates in Cu matrices and nanoscale Cu precipitates within Fe grains as observed experimentally in Figure 3.

With the in situ X-ray diffraction study, only reflections from Fe and Cu matrices are considered here because the reflections from nanoscale coherent precipitates cannot be reliably detected. Diffraction signal from matrix areas is at least one order of magnitude higher as compared to precipitates owing to the smaller volume fraction of Cu precipitates within the Fe matrix (i.e., ~10.1%); moreover, the fraction of Fe precipitates within the Cu matrix is even smaller, i.e., 1.7% [7]. Therefore, diffraction signal from matrix areas clearly dominates over the signal from the nanoscale precipitates. Even if reflections from Fe and Cu precipitates are present in diffraction patterns, it would be difficult to resolve due to lattice structure match of Cu matrix (i.e., FCC) with the precipitates of Fe (having FCC structure and being coherent with Cu matrix) and the precipitates of Cu (i.e., also FCC) within Fe grains [7].

Both Cu and Fe matrices produce arc-like diffraction lines with monochromatic beam indicating that these areas are nano-crystalline aggregates with essentially arbitrary orientations of nano-crystallites, although they exhibit preferred orientations; Figure 3d. X-ray diffraction patterns reported here have been obtained with X-ray beam focused down to 2–6 μm^2^, and, therefore, microstructural features in a scale of micrometers, e.g., both Cu and Fe grains, can be revealed by X-ray diffraction.

Phase transition from the BCC to HCP type of crystal lattice was clearly observed in Fe matrix during compression. During multiple X-ray diffraction experiments with monochromatic beam, the highest pressure when no Fe (HCP) phase was observed was 15.9 GPa (Figure 3e), and the lowest pressure when Fe (HCP) phase was reliably detected was 16.8 GPa, (Figure 3f). Therefore, onset of this transition was determined to occur in the range of 15.9 to 16.8 GPa, which is at least a few GPa higher than what is typically observed for pure Fe metal. In some of the reported studies on pure Fe metal [15,16,17,18,20,21,22], it was noted that the phase shift could also be altered by increasing the compression rate, and it will be valuable to further investigate this material under dynamic (shock type) compression and strain rates on the order of 10^5^/s.

Using Laue diffraction technique, it was observed that the 50Cu–50Fe alloy contains micron-sized single-crystals embedded into the nano-crystalline aggregates. Apart from the nano-crystalline areas, these crystals produced sharp diffraction spots; however, when monochromatic beam is used, these spots overlap with the arc-like diffraction lines from the nano-crystalline aggregates and make identification of these microcrystals extremely challenging. On the Laue diffraction images, the nano-crystalline areas produced only diffuse, extremely broad “streaky” reflections or just introduced rise to the background, while the microcrystals produced sharp and clearly distinguishable Laue diffraction spots.

Optical images of the sample before and after Laue diffraction measurements are presented in Figure 4a,b, respectively. The 2D translational Laue diffraction scan was collected on the sample first at pressure of 5.3 GPa, which was substantially below the onset of the BCC to HCP transition in the Fe matrix. The scan area is outlined as a black rectangle on the absorption scan across from the sample presented in Figure 4c. In order to observe alteration of the micro-crystals caused by further compression, a smaller area denoted by the dotted line on Figure 4c was selected. Series of 2D translation scans were collected on this area while increasing pressure. By indexation of the strongest Laue reflections from the scan area, four micro-crystals, referenced below as crystals 1–4, have been identified (Figure 5). Indices of these reflections match strong predicted reflections from Cu (FCC) lattice and do not match with the strongest predicted reflections from Fe (BCC) lattice. Therefore, all of these four crystals belong to Cu matrix with FCC type lattice. As presented in Figure 5a–c, mapping of reflections shows elongated shapes of crystals 1–3. Areas of all the four crystals 1–4 are outlined with respect to the scan area in Figure 5e. Crystals 1–3 are elongated in a parallel direction, attributed to the morphology of the Fe dendrites, which are also elongated parallel to each other (Figure 3a). The Cu phases located between the Fe matrix areas are also elongated in the same direction. The typical thickness of the Cu phase is in a range of few microns, which is consistent with the thicknesses of crystals 1–3. Crystal 4 has a more isometric shape, but it is located on edge of the sample, and the shape of the crystal 4 area is consistent with the suggestion that in the bulk sample this crystal had a similar shape to the other three crystals. The elongation of crystals 1–3 does not match to any specific crystallographic direction in their lattices. No regular orientation relationships between crystals 1–4 were identified. Crystals 1 and 2 are misoriented with respect to one another by an angle of 3.6°. All other crystals are misoriented by substantially larger angles.

The same Cu (FCC) microcrystals were identified also in bulk samples. As examples, indexed Laue diffraction patterns from microcrystals referenced below as crystals 5–9 are presented in Figure 6a–e along with maps of reflections. As presented in Figure 6f, crystals 5, 7 and 8 are located within the same area as samples cut by the laser system for high pressure studies. Therefore, the Cu (FCC) microcrystals were present in the DAC-sized samples originally and not due to recrystallization caused by heating during laser cutting of the sample. On the other hand, the Cu (FCC) microcrystals are also present in the areas located outside areas of laser-cut samples, e.g., crystal 9.

The Cu (FCC) microcrystals exhibit deformation across compression indicated by gradual broadening of their Laue reflections. As an example, in Figure 7a–d, maps of reflections from small parts of areas of crystals 1 and 2 are presented at 15.3 GPa and 16.3 GPa for comparison. All the presented reflections became more diffuse at the higher pressure, and it is important to note that this broadening is very homogeneous. Series of Laue diffraction patterns from crystals 1–4 and maps of reflections covering the entire areas of these crystals across compression are presented in Appendix A. As one can see from these videos, this deformation is somewhat inhomogeneous only in the sense that different parts of areas of crystals 1–4 may exhibit this deformation at slightly different pressures, but the type of deformation is essentially the same over the whole areas of crystals 1–4. Onset of this deformation is at very similar pressure to the onset of BCC to HCP lattice type transition in Fe matrix determined with monochromatic beam. Deformation of Fe dendrites associated with this transition may cause deformation of the microcrystals in Cu matrix as well, which would explain the coincidence of these pressures. However, in this case, deformation of the Cu (FCC) microcrystals would be more inhomogeneous. In other words, while some reflections still may exhibit broadening similar to the one presented in Figure 7a–d, other reflections would become more elongated and “streaky”. Furthermore, positions of Laue diffraction spots would exhibit larger variation across the sample area indicating an increase in intrinsic lattice rotation of crystals 1–4. Deformation of crystals 1–4 due to the BCC to HCP transition in the Fe matrix would increase lattice rotation causing substantial shifts of relative diffraction spot positions across crystals 1–4.

The observed type of deformation of crystals 1–4 is explainable by a transition of the Fe precipitates with FCC-type lattice causing deformation of the Cu matrix as well. As the precipitates are distributed homogeneously within the matrix area, it is reasonable to suppose that this transition of the precipitates takes place at similar pressures across the entire Cu matrix area, and, therefore, the deformation caused by the transition, as detected by the 2 μm^2^ X-ray beam, is also homogeneous. As was mentioned previously, different areas of crystals 1–4 may exhibit this deformation at slightly different pressures, but this is also explainable by the intrinsic inhomogeneity of crystals 1–4 in terms of density of dislocations and other defects of their lattices, which may affect the onset pressure of the transition of Fe (FCC) precipitates. It is interesting that onset pressures of the BCC to HCP transition of Fe dendrites and the suggested transition of Fe (FCC) precipitates coincide.

As presented in Appendix A after 16.8 GPa, crystals 1–4 produced only very diffused “streaks”. A 2D translation Laue diffraction scan was collected at this pressure over the whole area denoted by the black rectangle in Figure 4c. No sharp reflections from the sample have been observed. When diffraction measurements with monochromatic beam were conducted on this sample, pressure drifted up to 18.3 GPa. Broadened arc-like reflections from Cu (FCC) matrix have been observed with monochromatic beam approximately in the middle of the local scan area outlined by the dotted line in Figure 4c on which Laue data under compression were collected (Figure 8a). In total, 12 such reflections have been indexed with orientation matrix approximately matching to orientations of crystals 1 and 2. In Fe matrix, HCP lattice clearly dominated at this pressure, although BCC structure was also present. After decompression down to near ambient (~0.2 GPa), Fe matrix transformed back to BCC-type lattice without any notable changes in texture of Cu matrix (Figure 8b). A 2D Laue scan collected across the decompressed sample did not show any changes with respect to the highest-reached pressure.

### 3.2. Phase Transition at Nanometer Length Scales

The high-angle annular dark field technique in scanning transmission electron microscopy (HAADF-STEM) mode was used for the ex situ characterization of the structural changes that occurred in nano-precipitates (i.e., precipitates of Cu and Fe) during high-pressure experiments in 50Cu–50Fe alloy. A low-magnification HAADF micrograph shows the presence of Cu precipitates (indicated by yellow arrows) and Fe precipitates (indicated by red arrows) within Fe and Cu grains, respectively; Figure 9a. However, at higher magnification, modulated (or fringe type) structure can be seen within Cu precipitates (indicated by red arrows); Figure 9b.

One feature observed within Cu precipitates on numerous occasions is the change in local lattice configuration; Figure 9c. A squared region, **A** (~2.5 nm × 2.5 nm), within the center of the Cu precipitate exhibits a lattice structure similar to the BCC lattice (having a zone axis of <001>), as can be seen in the FFT diffractogram of the region (box **A**), which also consists of superlattice spots. Moreover, a region close to the interface with Fe grain (box **B**) shows FCC lattice structure (with zone axis of <011>), although the Fe grains were initially of BCC structure, whilst another region (box **C**) close to the Cu precipitate–Fe grain interface showed BCC lattice configuration (with zone axis of <001>), as confirmed from the corresponding FFT diffractogram; Figure 9c. In another case, the interface region between the Cu precipitate and Fe matrix exhibits the lattice structure of HCP (with zone axis of $\langle 2\bar{1}\bar{1}0\rangle$) as indicated by the corresponding FFT, whilst the interior of the precipitate still retains FCC lattice configuration (having zone axis of <110>); Figure 9d. One interesting feature observed within the Cu precipitate is a thin layer of region with a changed lattice arrangement from the surrounding FCC lattice of Cu. This region did not show configuration of any specific Bravais lattice; instead, it exhibits a faulted region of atomic stacking sequences; Figure 9e. This region showed the presence of different martensitic variants of Cu, such as 18H1; Figure 9e. This stacking-faulted region in Cu precipitates was found when <011> of FCC Cu was close to the <001> of BCC Fe; Figure 9e.

The Fe precipitates present in the Cu matrix have also undergone structural changes due to high pressure from their initial metastable FCC structure that was coherent with the adjacent Cu matrix. However, unlike Cu precipitates in Fe grains, Fe precipitates in Cu matrix showed only one kind of changed feature post high-pressured deformation. These Fe precipitates (diameter ~5 nm) showed regular-spaced modulated structure. However, FFTs acquired from these Fe-precipitated regions exhibit twin reflections in their corresponding diffraction patterns, which suggests these modulated structures within Fe precipitates primarily resulted from parallel-oriented multiple-twinned regions having ($01\bar{1})$ as twin planes, as can be observed in Figure 10a,b. Moreover, apart from formation of modulated structure within Fe precipitates, various regions of Cu matrix between the Fe precipitates exhibited modulated domains, as can be seen in Figure 10b. This modulation could be the possible reason for the observed deformation in the Cu matrix during Laue diffraction studies.

## 4. Discussion

Phase transition of FCC Cu into BCC Cu as observed in the current study has also been reported in earlier works [55,56,57,58]. A theoretical study showed that FCC crystal can be homogeneously deformed under uniaxial tensile load along the [100] direction, i.e., a path of minimum energy state which takes the crystal into a relaxed BCC configuration through the bifurcation path [58]. However, from the dynamic stability point of view, numerous theoretical studies had proposed that the BCC structure of Cu symbolized an unstable state where the bulk modulus either vanished or was even negative [35,59,60]. Therefore, it is difficult to find the BCC structure of pure Cu in bulk under common circumstances. Moreover, in computational studies on the Cu thin film deformation (i.e., in stretching or compression), formation of the complete BCC phase of Cu has hardly been observed, e.g., as a result of shock compression on Cu single crystal, the maximum volume fraction of BCC lattice configuration of single crystal Cu that could be achieved was 85% at 1.8 km/s shock velocity along [100] direction [61]. Similar to these studies, BCC lattice configuration of Cu precipitates have only been found in certain localized regions within Cu precipitates in the present study. No single Cu precipitate in its entire volume has been found to have undergone structural transition to BCC phase.

Previous research on phase transformations of Cu presented a detailed description of crystallographic mechanisms for FCC→BCC crystal structure transition and revealed the physical fundamentals of this phase transformation phenomenon [35,58]. Among these theories, the Bain mechanism has been found to be the most common theory for describing the phase transformation process of FCC→BCC Cu [47,48,49], accomplished by the shortest and simplest movement of atoms. During phase transformation of FCC Cu, deformation along one of the [100] directions results in reverse strain along the other two mutually orthogonal directions, i.e., along the [010] and [001] directions [62]. A recent study has indicated that the gradual increase in strain amount till 9.4% changes the lattice parameter along these [100] type directions marginally, while the applied strain of 12.4% significantly alters the lattice spacing along these directions, and it has been found that the initial lattice constant of FCC Cu (i.e., 0.3615 nm) changed to 0.286 nm along [100] direction in transformed BCC Cu lattice [62]. Hence, in the present study, the formation of BCC lattice configuration in regions within Cu precipitates possibly followed the similar crystallographic mechanism for transition from FCC to BCC structure, as can be seen for the BCC lattice structure along [100] direction in the FFT of Figure 9c.

Another feature of structural change that has been observed in Cu precipitates is the coexistence of FCC and HCP lattice configuration; Figure 9d. The dynamic process of FCC→HCP phase transformation in strained Cu films generally occurs through heterogeneous nucleation of HCP phase at the film’s surface (similar to the interface region of Cu precipitate–Fe grain in the present context; Figure 9d) and subsequently grows into the bulk [46]. However, in the current investigation, the HCP lattice configurations in Cu precipitates have only been observed close to their interfaces with the adjacent Fe grains (Figure 9d); hence, growth of this HCP phase to the interior of the Cu precipitates was absent, unlike those observed in Cu films [46]. Although most studies indicated FCC to HCP structural changes in Cu via BCC structure [62], in the current investigation, the presence of BCC structure adjacent to HCP has not been found in any instances. A possible reason for this could be the fact that the electron microscopy observations are intermittent in nature, while the computational studies are effective in showing the successive transformations of FCC→BCC→HCP phase transition in Cu [62]. In general, this phase transformation of FCC Cu to HCP Cu is mediated by plastic deformation. The structural evolution during the transformation is facilitated by the dislocation glide motion, and the transformed phase grows due to the propagation of the stacking faults bounded by the Shockley partial dislocations [46]. Previous studies stated the phase transformation of Cu from FCC→BCC [58], FCC→HCP [46] and FCC→BCC→HCP [62] lattice structures as martensitic in nature. However, owing to the inability to establish the mechanism of phase transformation with certainty, the structural transitions observed in the Cu precipitates in the current study have not been referred to as martensitic in nature.

A faulted region, found in Cu precipitate, cut across the entire precipitate’s longitudinal axis (Figure 9e) has a specific thickness (~1.5–1.8 nm) throughout its length. This fault placed along the closed packed (1$\bar{1}$1)_Cu_ planes can be found in the earlier observations of 9R structure in martensitic transformation of Cu precipitates [63]. This kind of faulted regions are caused from the faults in the stacking sequences of the closed packed planes, i.e., alteration in ABC arrangement of {111} planes for FCC. In the current example, the faults are observed in ABC sequences for $\left(\bar{1}\bar{1}1\right)$ planes, as can be seen from the enlarged view of the faulted layer in Figure 9e. This hexagonal type of faults in stacking sequence are a characteristic feature of 9R structure in Cu [64].

Structural changes occur in FCC Fe particles inside Cu matrix, i.e., a multiple-twinned-like structure within the particles occurs due to the deformation, and it starts when a glide dislocation interacts with a FCC Fe particle [31]. In the present study, it has been found that under 〈211〉 perpendicular beam conditions, both the primary slip plane $\left(1\bar{1}\bar{1}\right)$ and primary slip direction <$0\bar{1}\bar{1}>$ of FCC lattice are present for these particles; Figure 10a,b. This slip configuration facilitated the easy accumulation of shear strain by slip and resulted in the formation of elliptical shaped particles [31] having elliptical axes either along the slip directions or at an angle of 40°–45° to the primary slip direction; Figure 10a,b. This phenomenon suggests that the Fe particles were uniformly sheared or cut by dislocations during the high-pressure deformation [31]; hence, slip bands were hardly found in these particles. Subsequent formation of dislocation loops (e.g., Frank or Shockley) triggers the martensitic transformation of FCC Fe particles [65,66]. However, this might be essential but not sufficient criteria to start transformation in FCC Fe [66]. Further, these loops are found to grow by Shockley edge dislocations, while these propagate on other successive planes and lead to the formation of twins [65]. Moreover, atomistic study revealed that the twin boundaries are generally propagated by the glide of pairs of partial twin dislocations, while the propagation of FCC screw dislocations along the coherent interface (i.e., between FCC Fe particle and Cu matrix) plays an critical role for the subsequent lattice transformation of the Fe precipitates [65]. Owing to the ex situ nature of the present electron microscopy studies, coexistence of loops and twins in Fe particles have not been observed. However, in low-pressured (e.g., 10–12 GPa) samples, formation of dislocation loops around the Fe particles can be seen, in Figure 10c. This observation also suggests that the loop formation could be a precursor to the formation of twinned structure during FCC to BCC lattice transformation of Fe particles [65,66]. However, in the current investigation, the interface between Fe particles and the Cu matrix remains coherent with a change (~5–10%) in lattice spacing inside Fe precipitates.

## 5. Conclusions

Laser direct metal deposited Cu50Fe50 alloy was subjected to high-pressure deformation in a diamond anvil cell and the stability of the microstructure was characterized by in situ Laue diffraction study, whilst post deformation ex situ characterization was performed using transmission electron microscopy. Studies revealed phase transition of constituent phases (Cu and Fe) at multiscale, i.e., at microscale grains and at nanoscale precipitate domains, from their initial configurations. The onset of reversible phase transformation (BCC to HCP) in Fe at grain scale has been found to occur at approximately 16 GPa pressure. However, metastable FCC Fe nano-precipitates within the Cu grains show a modulated structure in the form of multiple nano-twinned (with {011} type twin planes) domains within the precipitates. TEM studies exhibit different modes of structural changes in FCC Cu precipitates due to high-pressure deformation, such as local alterations in lattice structures (e.g., development of regions with metastable BCC and HCP lattice) from its initial FCC lattice and the formation of large stacking faults across the precipitates.

## Figures and Tables

**Figure 1 nanomaterials-12-01514-f001:**
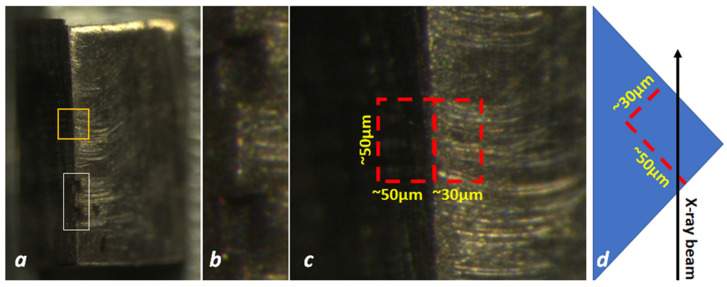
(**a**) Bulk sample of 50Cu–50Fe. (**b**) Cavities formed as a result of samples being cut by the laser cutting machine (white box in (**a**)) for high pressure measurements. (**c**) Laue diffraction scan area (yellow box in (**a**)) as projected onto horizontal and vertical planes in (**d**). Red dotted lines denote projections of edges of a DAC-sized sample that could be cut by the laser system from the scan area.

**Figure 2 nanomaterials-12-01514-f002:**
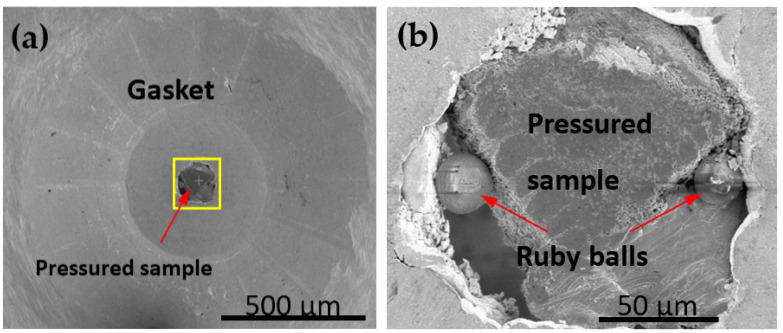
Samples used for ex situ microscopy studies after withdrawing these from high pressure tests. (**a**) Specimen placed inside the membrane as it was during high pressure experiments. (**b**) Enlarged view of yellow box region in (**a**) showing the arrangement of sample supported by Ruby balls in either side to hold it inside the gasket hole.

**Figure 3 nanomaterials-12-01514-f003:**
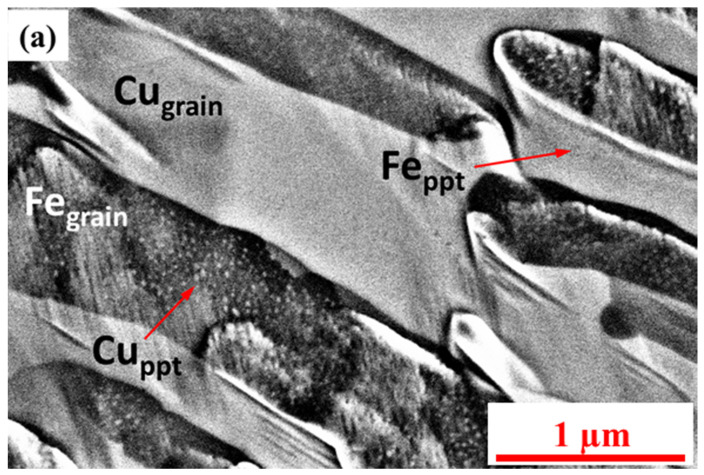
(**a**) SEM micrograph showing two phase-separated regions of Cu and Fe (i.e., Cu_grain_ and Fe_grain_) and the presence of precipitates (i.e., Cu_ppt_ and Fe_ppt_) formed during the laser DMD process. HAADF-STEM micrographs exhibiting (**b**) coherent Fe precipitate (FCC structure) in Cu matrix and (**c**) Cu precipitate (FCC structure) in Fe grain. (**d**) Diffraction patterns obtained on Cu and Fe matrices of Cu50Fe50 precipitates with X-ray monochromatic beam at 16BMD beamline of APS. The left-hand-side pattern indicates a beam size of 5 × 3.5 μm^2^ with no sample rotation, while the right-hand-side pattern indicates a beam size of 4 × 4.5 μm^2^ with sample rotated by 50°. Diffraction patterns showing (**e**) no presence of Fe (HCP) at 15.9 GPa pressure, (**f**) and the presence of Fe (HCP) at 16.8 GPa.

**Figure 4 nanomaterials-12-01514-f004:**
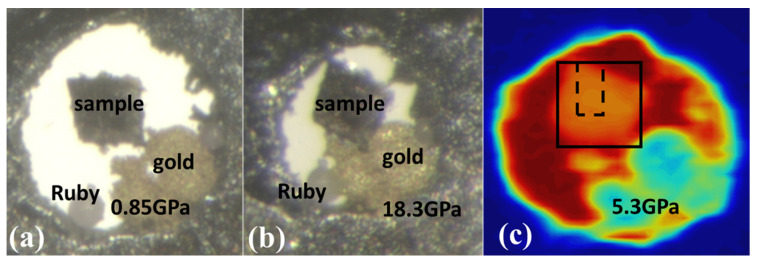
Optical images of the sample in DACs before (**a**) and after (**b**) in situ Laue diffraction measurements. Absorption scan across the sample (**c**). Black rectangle denotes the Laue diffraction scan area, and dotted line outlines the local scan area—which is explained further in the text. Size of the Laue diffraction scan area is 50 × 50 μm^2^.

**Figure 5 nanomaterials-12-01514-f005:**
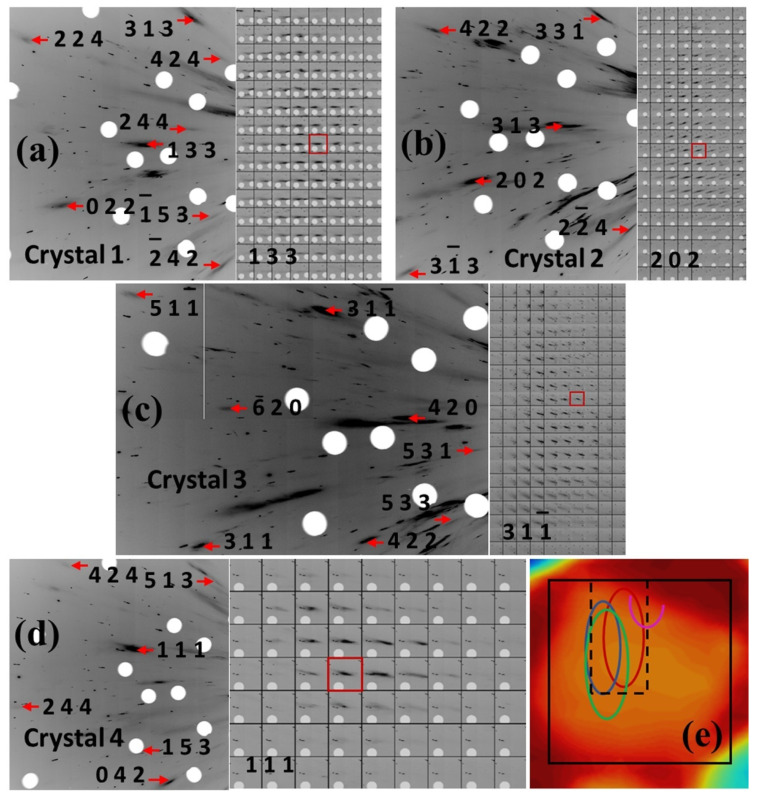
(**a**–**d**) Laue diffraction patterns (left) and maps of respective reflections (right) from crystals 1–4 obtained at 5.3 GPa. Red rectangles denote the position where diffraction patterns were collected. (**e**) Encircled areas of crystals 1 (red), 2 (blue), 3 (green) and 4 (magenta). Black rectangle denotes the Laue diffraction scan area, and dotted line outlines the local scan area—which is explained further in the text.

**Figure 6 nanomaterials-12-01514-f006:**
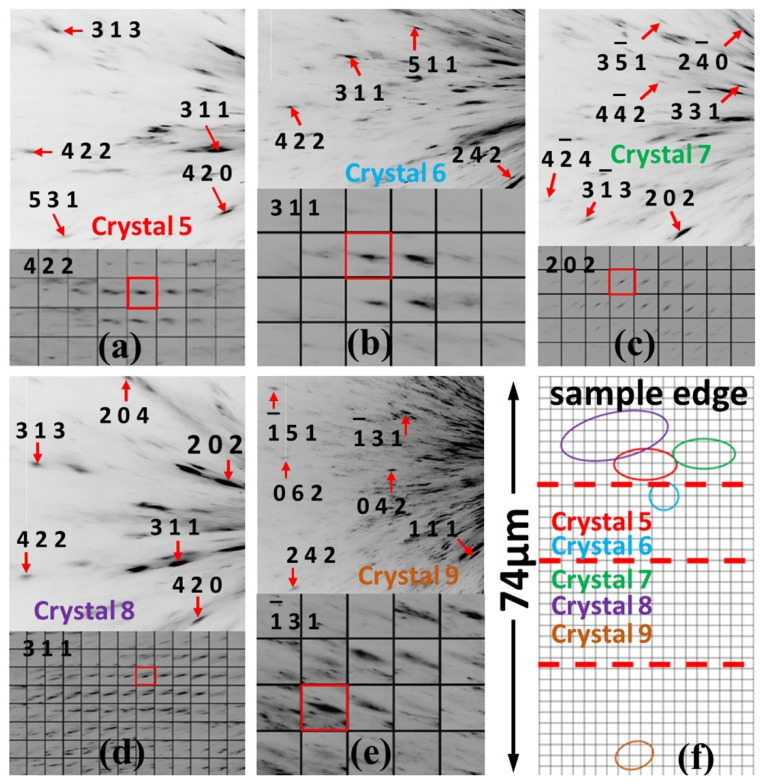
(**a**–**e**) Laue diffraction patterns (top) and maps of the respective reflections (bottom) obtained from crystals 5–9 in the bulk sample. Red rectangles denote positions where diffraction patterns in crystals were collected. (**f**) Laue diffraction scan area within the bulk sample as projected onto the plane perpendicular to the incident X-ray beam. Ovals denote areas of crystals 5–9. Red dotted lines denote projections of edges of a DAC-sized sample that could be cut by the laser system from the scan area.

**Figure 7 nanomaterials-12-01514-f007:**
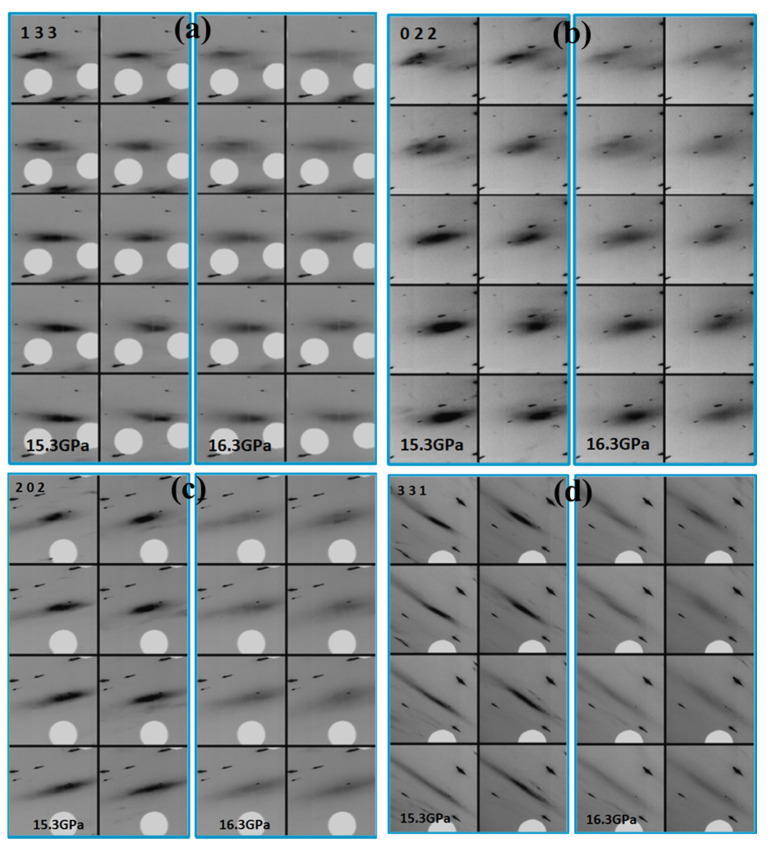
Maps of Laue reflections from crystal 1 (**a**,**b**) and crystal 2 (**c**,**d**) showing pressure-induced broadening. The intensity scaling is same as defined in Fit2d program [54].

**Figure 8 nanomaterials-12-01514-f008:**
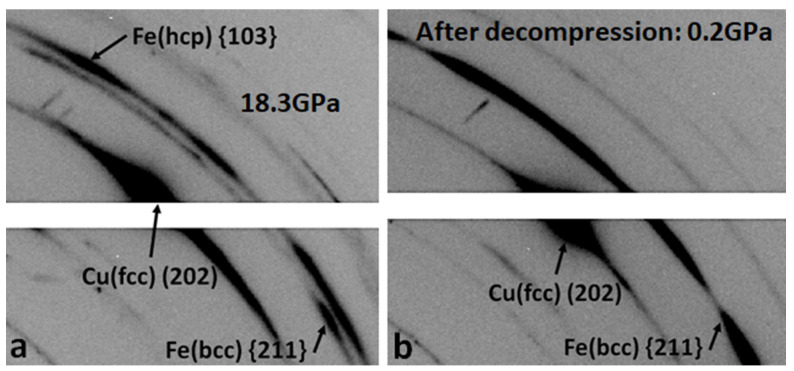
Diffraction patterns obtained on Cu and Fe matrices of Cu50Fe50 precipitates with X-ray monochromatic beam at 16IDB beamline of APS. Beam size 3 × 6 μm^2^; sample was rotated by 1°.

**Figure 9 nanomaterials-12-01514-f009:**
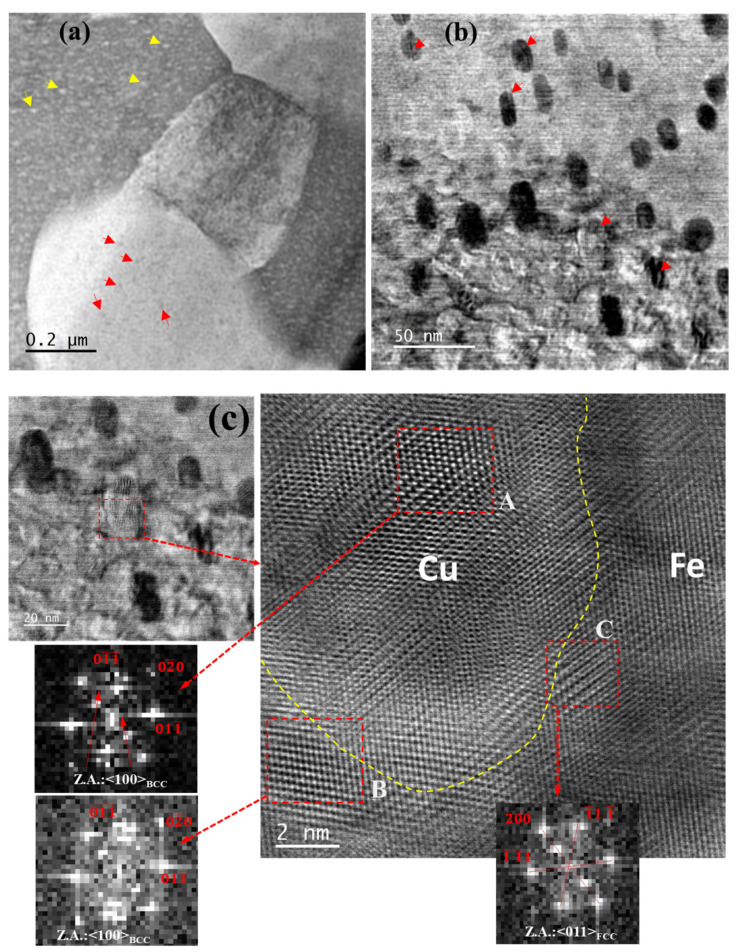
In post high-pressured (i.e., at 18.3 GPa) deformed samples: (**a**) Low magnification STEM-HAADF image showing presence of Cu and Fe precipitates in Fe and Cu grains, respectively. (**b**) Relatively higher magnification reveals fringe type characteristic (or modulated structure) in Cu precipitates. One kind of structural change observed in Cu precipitates exhibiting (**c**) changes in lattice structure from stable FCC to BCC at local regions, (**d**) formation of HCP lattice configuration and (**e**) presence of stacking faults within Cu precipitate.

**Figure 10 nanomaterials-12-01514-f010:**
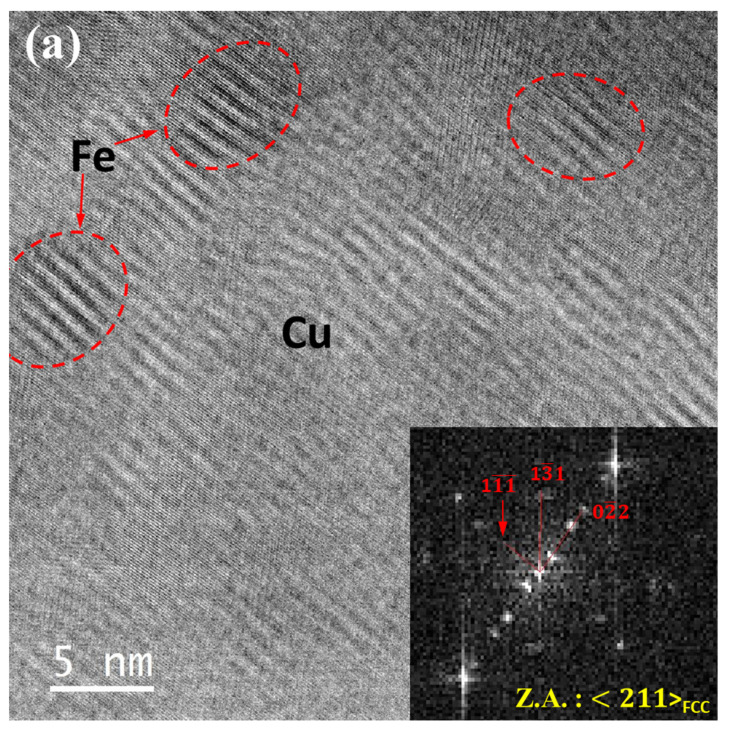
(**a**) Modulated and multiple twinned structures of Fe precipitates (encircled by red dotted lines) in Cu grains after high pressure (i.e., at 18.3 GPa) deformation. (**b**) Apart from modulated Fe precipitates, Cu matrix regions with modulated configuration. (**c**) Formation of dislocation loops around Fe precipitates at stress level (i.e., at 11.3 GPa) below the onset of phase transformation of Fe.

## Data Availability

The data presented in this study are available on request from the corresponding author.

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
