# Peer review of "Phase Transitions of Cu and Fe at Multiscales in an Additively Manufactured Cu–Fe Alloy under High-Pressure"

_nanomaterials, 2022, doi:10.3390/nano12091514_

Round 1

Reviewer 1 Report

Dear Dr. Chatterjee,

your latest publication "Phase transitions of Cu and Fe at multiscales in
additively manufactures Cu-Fe Alloy under high-Pressure" represents a in-depth phase analysis. Although this article deserves publication in essence, I have some aspects that you may wish to consider before publication.

  • Please explain in more detail why you are sure that Fe(bcc) to Fe(hcp)
    represents a martensitic phase transformation - although you state that there is a steady discussion that this phase transformation can also be pressure induced. In the following (page 12) you declare the transition in Cu fcc->bcc->hcp to be of martensitic nature. To my understanding this is not possible. Please explain in detail how this twofold phase transition would happen if being of martensitic nature.

  • Please explain wyh such a complicated system is under investigation. Why do you need to apply DMD, why is there a need for investigation of the alloy? What would be expected if pure Cu and Fe would be treated a similar manner? What is the benefit of investigating the phases in the vicinity of the other one?

  • The section 2.1 is much to short. The alloy is not single phase and the phase formation and distribution will strongly depend on the mixture of the powder, their particle size and distribution, the volume of the melt, as well as the cooling rate during crystallisation. All these aspects require a sound documentation and discussion.

  • I do not understand Fig. 1e. Further, please also give indications for crystals 1-4 as they are later referred to in the text.

  • Please modify your images in such a way that no text (labels etc) is smaller than the regular text, when the images are scaled to their final size, e.g. scaling bars in Fig. 3 are hardly readable. Please avoid using yellow colour in the labels in the images as in particular Fig. 6 becomes hard to read.

  • page 10: Would it be possible to separate information arising from matrix and precipitates?

  • page 17. "in situ" is the wrong term here and must not be used in this context.

  • page 17 last paragraph before the conclusions: This paragraph should be explained in more detail.

  • Regarding the author contributions, I have to admit, that "supervision",
    "project administration" and "funding acquisition" do not justify a
    co-authorship. As "Conceptualization" is not explained, I suppose that this is nothing more than writing the grant proposal while thinking about possible scientific questions. Thus, the authors N.V. and A.M. contributed to the article by proof-reading, only. To my understanding, this does not justify a co-authorship. Please explain in detail what these authors contributed to the present article (and not to the project).

Reviewer 2 Report

After reading the abstract, I thought that the research focus of the article might be liquid-liquid phase separation owing to the high cooling rate during additive manufacturing and the special Cu-Fe system. According to my knowledge, Fe and Cu are miscible as liquids but immiscible in the solid state. The microstructure of the printed parts significantly depends on the cooling rate and the initial composition owing to spinodal, binodal decomposition and undercooling. Nevertheless, the liquid-liquid phase separation was not discussed in the article.

As I am not an expert in the realm of “Laue diffraction”, it is hard for me to judge whether this article is worth publishing or not. Nevertheless, the following points should be elaborated.

  1. Page 2: The author mentioned the range of particle diameter. However, it is recommended to show particle size distribution like d10, d50, and d90.
  2. The authors should clarify why additive manufacturing has been used in the work; what are the advantages of additive manufacturing compared to conventional (vacuum) casting or powder metallurgical methods?
  3. The motivation is unclear to me. The stress-induced phase transition can be observed in different material systems. Could this phenomenon observed in the Cu-Fe system be used for certain applications or important for material science?
  4. The interpenetrated microstructure shown in Fig. 3 is a common feature after spinodal decomposition. The formation of the tiny spherical inclusions might be caused by the secondary liquid-liquid phase separation (binodal decomposition of undercooled Cu-Fe liquid) or the precipitation reaction in the solid state. The mechanism of the formation of the nano-twinned structures with small inclusions should be elaborated.

Round 2

Reviewer 2 Report

1.    What is the meaning of “d5–d75 µm” and “d25—d150 µm”?
2.    As stated by the authors, the unique nature of DMD is the high cooling rate, which can from my point of view result in high undercooling and spinodal or binodal decomposition of the Cu-Fe system. In my opinion, to elaborate on the microstructure features derived by means of DMD, liquid-liquid phase separation should be discussed.
